# SCoRe: Submodular Combinatorial Representation Learning for Real-World Class-Imbalanced Settings

## Abstract

Representation Learning in real-world class-imbalanced settings has emerged as a challenging task in the evolution of deep learning. Lack of diversity in visual and structural features for rare classes restricts modern neural networks to learn discriminative feature clusters. This manifests in the form of large inter-class bias between rare object classes and elevated intra-class variance among abundant classes in the dataset. Although deep metric learning approaches have shown promise in this domain, significant improvements need to be made to overcome the challenges associated with class-imbalance in mission critical tasks like autonomous navigation and medical diagnostics. Set-based combinatorial functions like Submodular Information Measures exhibit properties that allow them to simultaneously model diversity and cooperation among feature clusters. In this paper, we introduce the **SCoRe**[1] (**S**ubmodular **Co**mbinatorial **Re**presentation Learning) framework and propose a family of Submodular Combinatorial Loss functions to overcome these pitfalls in contrastive learning. We also show that existing contrastive learning approaches are either submodular or can be re-formulated to create their submodular counterparts. We conduct experiments on the newly introduced family of combinatorial objectives on two image classification benchmarks - pathologically imbalanced CIFAR-10, subsets of MedMNIST and two real-world object detection benchmarks - India Driving Dataset (IDD) and LVIS (v1.0). Our experiments clearly show that the newly introduced objectives like Facility Location, Graph-Cut and Log Determinant outperform state-of-the-art metric learners by up to 7.6% for the imbalanced classification tasks and up to 19.4% for object detection tasks.

## 1 Introduction

Deep Learning models (Krizhevsky et al., 2012; He et al., 2016; Simonyan & Zisserman, 2015) for representation learning tasks distinguish between object classes by learning discriminative feature embeddings for each class in the training dataset. Most State-of-the-Art (SoTA) approaches adopt Cross-Entropy (CE) (Baum & Wilczek, 1987) loss as the objective function to train models on cannonical benchmarks. In contrast to curated cannonical benchmarks which present a balanced data distribution, real-world, safety critical tasks like autonomous navigation, health-care etc. demonstrate class-imbalanced settings. This introduces large inter-class bias and intra-class variance (Li & Wang, 2020) between object classes while training a deep learning model which CE loss is unable to overcome.

Recent developments in this field have observed an increase in the adoption of deep metric learning approaches (Ranasinghe et al., 2021; Deng et al., 2019; Wang et al., 2018) and contrastive learning (Schroff et al., 2015; Sohn, 2016; Khosla et al., 2020) strategies. In both supervised (Khosla et al., 2020; Ranasinghe et al., 2021) and unsupervised (Chen et al., 2020a) learning settings, these techniques operate on image pairs, rewarding pairs with the same class label ($positive$) to be closer while pairs from different class labels ($negative$) to be farther away in the feature space. Unfortunately, these approaches use pairwise metrics in their objectives which cannot guarantee the formation of tight or disjoint feature clusters in real-world settings. The experiments conducted by us in this paper

---

[1]The code for the proposed framework has been released at `https://anonymous.4open.science/r/SCoRe-8DE5/`.

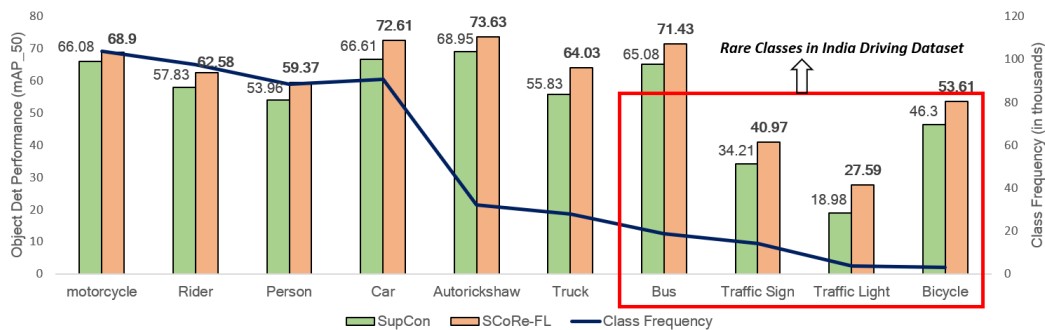

Figure 1: **The effect of class-imbalance** on the performance metrics ($mAP_{50}$) for the object detection task of the India Driving Dataset (IDD). As the class frequency (shown as blue line) decreases, proposed objectives in SCoRe (shown in red) consistently outperforms SoTA approaches like SupCon (shown in green) in detecting rare road objects like *bicycle*, *traffic light* etc. in IDD.

show that the aforementioned pitfalls lead to poorer performance of these techniques in real-world, class-imbalanced settings. This requires us to study metric learners from a combinatorial point of view by considering class specific feature vectors as sets and employing objectives that jointly model inter-class separation and intra-class compactness. Generalization of set-based information-theoretic functions like entropy, facility-location etc., also known as submodular functions (Fujishige, 2005b) have been shown Kaushal et al. (2019) to be effective in modeling diversity, representation, coverage, and relevance among sets in various machine learning tasks like active learning (Kothawade et al., 2022a; 2021) and subset selection (Killamsetty et al., 2021; Karanam et al., 2022).

In this paper, we propose a framework - **Submodular Combinatorial Representation** Learning (**SCoRe**) - that introduces set-based submodular combinatorial loss functions to overcome the challenges in class-imbalanced settings as shown in Figure 1. We propose a family of objective functions based on popular submodular functions Iyer et al. (2022) such as Graph-Cut, Facility Location and Log-Determinant. These set-based loss functions maximize the mutual information Kothawade et al. (2022b) between class-specific sets while preserving the most discriminative features for each set (class) thereby reducing the effect of inter-class bias and intra-class variance. Our results in Section 4 show that our proposed loss functions consistently outperform SoTA approaches in metric learning for class-imbalanced classification tasks for CIFAR-10 and MedMNIST Yang et al. (2023) datasets alongside object detection in real-world unconstrained environments like in India Driving Dataset (IDD) Varma et al. (2019) and LVIS Gupta et al. (2019). The main contributions of this paper can be summarized below:

- We introduce a novel submodular combinatorial framework of loss functions (SCoRe) which demonstrates resilience against class-imbalance for representation learning while outperforming SoTA approaches by up to 7.6% for classification task and 19.4% for object detection tasks.

- We highlight that existing contrastive learning objectives are either submodular or can be reformulated as submodular functions. We show that the submodular counterparts of these contrastive losses outperform their non-submodular counterparts by 3.5 - 4.1%.

- Finally, we demonstrate the performance of our proposed objectives for road object detection in real-world class-imbalanced setting like IDD (Varma et al., 2019) and LVIS Gupta et al. (2019) to show improvements up to 19.4% proving the reliability of our approach in real-world downstream tasks.

## 2 RELATED WORK

**Longtail Learning:** Visual recognition tasks in longtail learning are characterized by learning representative feature sets from a few over-represented classes (head) and many under-represented ones (tail). Traditional approaches in this domain have addressed the class-imbalance in longtail datasets by either over-sampling the rare (tail) classes (Chawla et al., 2002) or under-sampling the abundant (head) classes (Cui et al., 2019; Zhang et al., 2021). Both these techniques alter the distribution of the original dataset leading to poor generalization of the trained model. Alternatively, approaches like (Shu et al., 2019; Wang et al., 2017; Zhou et al., 2020) have resorted to re-weighting individual

class probabilities during representation learning which weight tail classes higher or disallow gradient updates for abundant classes (Tan et al., 2020). Recent works in this domain Zhou et al. (2020) shows that re-weighting and re-weighting strategies proposed in result in poor representation learning abilities. Further advancements in this field has been achieved by introducing metric / contrastive learning strategies in model training (Cui et al., 2021; 2023; Li et al., 2021; Zhu et al., 2022). Popularly adopted objective like (Khosla et al., 2020; Chen et al., 2020a) employ a two stage training strategy requiring large number of negative label information alongside large batch sizes. This makes them cumbersome to train on large datasets like ImageNet-LT Deng et al. (2009). Contextual feature learning through vision transformers Dosovitskiy et al. (2021) coupled with aforementioned techniques (Tian et al., 2022; Iscen et al., 2023) have also demonstrated huge performance gains at the cost of large compute resources. Very recently, approaches like Cui et al. (2021; 2023) and Du et al. (2023) have adopted a combination of data-augmentation and contrastive learning to achieve SoTA in long tail learning. While GPaCo introduces parametric learnable class centres by adopting representation learner as in Chen et al. (2020b), GLMC (Du et al., 2023) proposes a global and local mixture consistency loss, generated through global MixUp (Zhang et al., 2018) and local CutMix (Yun et al., 2019) techniques, and a cumulative head-tail soft label reweighted loss. Surprisingly, all SoTA approaches have adopted a distinct flavor of contrastive learning in their model training which signals the need for a generalizable contrastive learning strategy for representation learning.

**Metric and Contrastive Learning:** Traditional models trained using Cross-Entropy (CE) loss (Rumelhart et al., 1986) are not robust to class-imbalance, noisy labels etc. Approaches in supervised learning adopt metric learning (Deng et al., 2019; Wang et al., 2018; Ranasinghe et al., 2021; Wang et al., 2019) which learns a distance (Schroff et al., 2015) or a similarity (Deng et al., 2019; Wang et al., 2018) metric to enforce orthogonality in the feature space (Ranasinghe et al., 2021) while learning discriminative class-specific features. Another branch in representation learning, also known as contrastive learning, stems from noise contrastive estimation (Gutmann & Hyvärinen, 2010) and is popular in self-supervised learning (Chen et al., 2020a; He et al., 2020; Chen et al., 2020b) where no label information is available during model training. Application of these approaches in the supervised learning domain by SupCon (Khosla et al., 2020) aim to learn discriminative feature clusters rather than aligning features to their cluster centroids. Triplet loss (Schroff et al., 2015) uses only 1 positive and negative pair. N-pairs (Sohn, 2016) loss contrasts 1 positive against multiple negative pairs while SupCon contrasts between multiple positive and negative pairs. Lifted-Structure loss (Song et al., 2016) contrasts the similarity between positive image pairs and hardest negative pair. Additionally, SupCon bears close resemblance to Soft-Nearest Neighbors loss (Frosst et al., 2019) and maximizes the entanglements between classes. Although, these methods have demonstrated great success, they continue to adopt pairwise similarity metrics and are thus cannot guarantee formation of disjoint clusters.

**Submodular Functions:** Submodular functions are set functions that satisfy a natural diminishing returns property. A set function $f : 2^V \to \mathbb{R}$ (on a ground-set $V$) is submodular if it satisfies $f(X) + f(Y) \geq f(X \cup Y) + f(X \cap Y), \forall X, Y \subseteq V$ (Fujishige, 2005a). These functions have been studied extensively in the context of data subset selection (Kothawade et al., 2022b) , active learning (Kothawade et al., 2022a) and video-summarization (Kaushal et al., 2019; Kothawade et al., 2022b). Submodular functions are capable of modelling diversity, relevance, set-cover etc. which allows them to discriminate between different classes or slices of data while ensuring the preservation of most relevant features in each set. Very recent developments in the field have applied submodular functions like Facility-Location in metric learning (Oh Song et al., 2017). These properties of submodular functions can be used to learn diverse feature clusters in representation learning tasks which is a field yet to be studied in literature.

To the best of our knowledge , we are the first to demonstrate that combinatorial objectives using submodular functions are superior in creating tighter and well-separated feature clusters for representation learning. We are fore-runners in showing through Section 4 that most of the existing contrastive learning approaches have a submodular variant which have shown to outperform their non-submodular counterparts.

## 3 METHOD

In this section we describe the various components of our framework - SCoRe , for supervised representation learning tasks. Our framework is structurally similar to Khosla et al. (2020) and Chen et al. (2020a) with modifications to contrast existing metric learning approaches against submodular combinatorial objectives.

### 3.1 SCoRe: Submodular Combinatorial Representation Learning Framework

Supervised training for representation learning tasks proceed with learning a feature extractor $F(I, \theta)$ followed by a classifier $Clf(F(I, \theta))$ which categorizes an input image $I$ into its corresponding class label $c_i$, where $c_i \in [1, 2, ...C]$. Unlike standard model training for image classification , our supervised learning framework consists of three major components and is trained using a two-stage training strategy as introduced in Khosla et al. (2020) : (1) *Feature Extractor* , $f(I, \theta)$ is a convolutional neural network (Krizhevsky et al., 2012; Simonyan & Zisserman, 2015; He et al., 2016) which projects an input image $I$ into a $D_f$ dimensional feature space $r$, where $r = F(I, \theta) \in \mathbb{R}^{D_f}$, given parameters $\theta$.

In this paper we adopt residual networks (He et al., 2016) (specifically ResNet-50) as our feature extractor and aim to learn its parameters $\theta$ using an objective function described below. (2) *Classifier* , $Clf(r, \theta)$ is a linear projection layer that projects the $D_f$ dimensional input features $r$ to a smaller $D_p$ dimensional vector $z$, where $z = Clf(r, \theta) \in \mathbb{R}^{D_p}$ such that a linear classifier can classify the input image $I$ to its corresponding class label $c_i$ for $i \in [1, C]$. (3) *Combintorial Loss Functions* , $L(\theta)$ trains the feature extractor $F$ over all classes $C$ in the dataset to discriminate between classes in a multi-class classification setting. Contrastive and combinatorial losses largely depend on feature distance $D_{ij}(\theta)$ or similarity $S_{ij}(\theta)$ between pairs/sets $i$ and $j$ to compute the loss metric which depends on $\theta$. By varying the objective function for a given metric

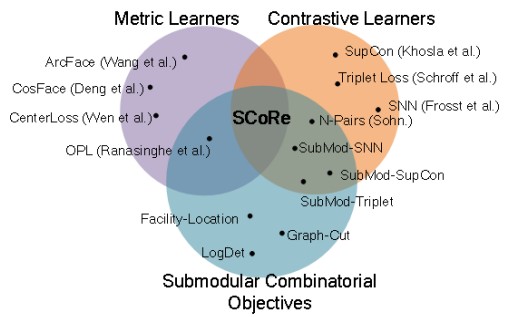

Figure 2: **Overview of Combinatorial Objectives** in SCoRe with respect to contrastive and metric learners.

, we are able to study their behavior in learning discriminative feature sets for each class in $\mathcal{T}$.

Training and evaluation of models using this proposed framework occurs in two stages. In stage 1 we train a generalizable feature extractor $F(I, \theta)$ using multiple variants of $L(\theta)$ on a large scale image dataset $\mathcal{T}$ containing $\{x_i, y_i\}_{i=1,2,...|\mathcal{T}|}$. In stage 2, we freeze the feature extractor $F(I, \theta)$ and train only a linear classifier $Clf(r, \theta)$ on the embeddings ($r$) generated by the feature extractor, using the standard cross-entropy loss Rumelhart et al. (1986). Using this framework we also propose a novel family of combinatorial objective functions which is formulated as the sum of submodular functions Iyer et al. (2022) and is discussed in Section 3.2.

### 3.2 Combinatorial Loss Functions

As introduced in Section 3.1 we propose a set of combinatorial loss functions which promotes the learning of tighter and well-separated feature clusters in the embedding space. This family of objectives considers each class $c_k$ in the dataset $\mathcal{T}$ as a set $A_k$ where $k \in [1, C]$. The task of $f$ is to learn the model parameters $\theta$ using $L(\theta)$ to enforce sufficient decision boundaries between the feature clusters while rewarding the formation of tighter clusters for each class $A_k$. The overall loss $L(\theta)$ can be defined as the sum over the loss $L(\theta, A_k)$ calculated for each set $A_k$ in the dataset, $L(\theta) = \sum_{k=1}^{|C|} L(\theta, A_k)$. Given multiple sets $A_1, A_2, A_3, ....A_{|C|}$ (each set contains instances of a single class) and a ground set $\mathcal{V}$ (which is the entire dataset), and a submodular function $f$, different formulations of combinatorial information measures Fujishige (2005a) can be defined. Define the Total Submodular Information as: $S_f(A_1, A_2, A_3, \ldots, A_{|C|}) = \sum_{k=1}^{|C|} f(A_k)$. Also, we can define the Total Submodular Correlation as: $C_f(A_1, A_2, A_3, \ldots, A_{|C|}) = \sum_{k=1}^{|C|} f(A_k) - f(\bigcup_{k=1}^{|C|} A_k)$. Given any submodular function, we can define two variants of combinatorial loss functions:

$$L_{S_f}(\theta) = S_f(A_1, \cdots, A_{|C|}), \;\; L_{C_f}(\theta) = C_f(A_1, \cdots, A_{|C|}) \tag{1}$$

The functions that we consider in this work, are defined with similarity kernels $S$, which in turn depend on the parameters $\theta$. A loss function which minimizes $S_f$ maximizes the intra-cluster similarity (by minimizing the submodular function on each cluster), while the loss function that

Table 1: Summary of various objective functions studied through SCoRe framework and their respective combinatorial properties (detailed derivations in section A.5 of the appendix).

| Objective Function | Equation $L(\theta, A_k)$ | Combinatorial Property |
|---|---|---|
| Triplet Loss Schroff et al. (2015) | $L(\theta, A_k) = \sum_{\substack{i,p \in A_k \\ n \in \mathcal{V} \setminus A_k}} max(0, D_{ip}^2(\theta) - D_{in}^2(\theta) + \epsilon)$ | Not Submodular |
| N-Pairs Loss Sohn (2016) | $L(\theta, A_k) = -[\sum_{i,j \in A_k} S_{ij}(\theta) + \sum_{i \in A_k} log(\sum_{j \in \mathcal{V}} S_{ij}(\theta) - 1)]$ | Submodular |
| OPL Ranasinghe et al. (2021) | $L(\theta, A_k) = (1 - \sum_{i,j \in A_k} S_{ij}(\theta)) + (\sum_{i \in A_k} \sum_{j \in \mathcal{V} \setminus A_k} S_{ij}(\theta))$ | Submodular |
| SNN Frosst et al. (2019) | $L(\theta, A_k) = -\sum_{i \in A_k}[\log \sum_{j \in A_k} \exp(S_{ij}(\theta)) - \log \sum_{j \in \mathcal{V} \setminus A_k} \exp(S_{ij}(\theta))]$ | Not Submodular |
| SupCon Khosla et al. (2020) | $L(\theta, A_k) = [\frac{-1}{|A_k|} \sum_{i,j \in A_k} S_{ij}(\theta)] + \sum_{i \in A_k}[log(\sum_{j \in \mathcal{V}} S_{ij}(\theta) - 1)]$ | Not Submodular |
| Submod-Triplet | $L(\theta, A_k) = \sum_{\substack{i \in A_k \\ n \in \mathcal{V} \setminus A_k}} S_{in}^2(\theta) - \sum_{i,p \in A} S_{ip}^2(\theta)$ | Submodular |
| Submod-SNN | $L(\theta, A_k) = \sum_{i \in A_k}[\log \sum_{j \in A_k} \exp(D_{ij}(\theta)) + \log \sum_{j \in \mathcal{V} \setminus A_k} \exp(S_{ij}(\theta))]$ | Submodular |
| Submod-SupCon | $L(\theta, A_k) = -[\sum_{i,j \in A_k} S_{ij}(\theta)] + \sum_{i \in A_k}[log(\sum_{j \in \mathcal{V} \setminus A_k} \exp(S_{ij}(\theta)))]$ | Submodular |
| Graph-Cut $[S_f]$ (ours) | $L(\theta, A_k) = \sum_{i \in A_i} \sum_{j \in \mathcal{V} \setminus A_k} S_{ij}(\theta) - \lambda \sum_{i,j \in A_k} S_{ij}(\theta)$ | Submodular |
| Graph-Cut $[C_f]$ (ours) | $L_{C_f}(\theta, A_k) = \lambda \sum_{i \in A_i} \sum_{j \in \mathcal{V} \setminus A_k} S_{ij}(\theta)$ | Submodular |
| Log-Determinant $[S_f]$ (ours) | $L(\theta, A_k) = \log \det(S_{A_k}(\theta) + \lambda \mathbb{I}_{|A_k|})$ | Submodular |
| Log-Determinant $[C_f]$ (ours) | $L(\theta, A_k) = \log \det(S_{A_k}(\theta) + \lambda \mathbb{I}_{|A_k|}) - \log \det(S_{\mathcal{V}}(\theta) + \lambda \mathbb{I}_{|\mathcal{V}|})$ | Submodular |
| Facility-Location $[C_f / S_f]$ (ours) | $L(\theta, A_k) = \sum_{i \in \mathcal{V} \setminus A_k} max_{j \in A_k} S_{ij}(\theta)$ | Submodular |

minimizes $C_f$ trades-off between maximizing the intra-cluster similarity but also minimizes the inter-cluster similarity (by maximizing $f(\cup_k A_k)$).We select the best of these information-theoretic formulations effectively in our framework to design a family of objective functions as shown in Table 1 for representation learning tasks.

### 3.2.1 SCoRe: Submodular Combinatorial Loss Functions

In this paper we propose three novel objective functions based on submodular information measures : Facility-Location (FL), Graph-Cut (GC), and Log Determinant (LogDet) as $L(\theta, A_k)$ and minimize them to overcome inter-class bias and intra-class variance. We adopt the cosine similarity metric $S_{ij}(\theta)$ as used in SupCon Khosla et al. (2020) which can be defined as $S_{ij}(\theta) = \frac{F(I_i, \theta)^\mathrm{T} \cdot F(I_j, \theta)}{||F(I_i, \theta)|| \cdot ||F(I_j, \theta)||}$. For objective functions which adopt a distance metric $D_{ij}$ we adopt the euclidean distance as in Schroff et al. (2015).

**Facility Location** (FL) based objective function minimizes the maximum similarity $S$ between sets of features belonging to different classes, $S_{ij}$ where $i \neq j$. The equation describing this objective is shown in Equation 2. This objective function ensures that the the sets $A_k \in \mathcal{V}$ are disjoint by minimizing the similarity between features in $A_k$ and the hardest negative feature vectors in $\mathcal{V} \setminus A_k$. Inherently, this function also learns the cluster centroid Fujishige (2005a); Iyer et al. (2022) for each $A_k$ in the embedding space thus proving to be effective in overcoming inter-cluster bias in downstream tasks.

$$L_{S_f}(\theta, A_k) = \sum_{i \in \mathcal{V} \setminus A_k} max_{j \in A_k} S_{ij}(\theta) + |\mathcal{V}|, \quad L_{C_f}(\theta, A_k) = \sum_{i \in \mathcal{V} \setminus A_k} max_{j \in A_k} S_{ij}(\theta) \quad (2)$$

Note that in the case of FL, $L_{S_f}$ and $L_{C_f}$ differ by a constant, and are the same loss. We also point out that this loss function naturally handles the cases where the classes are imbalanced: it boosts the imbalanced classes since $\mathcal{V} \setminus A_k$ is actually going to be larger for imbalanced classes compared to the more frequent classes. As expected, the FL loss performs the best in imbalanced data settings.

**Graph Cut** (GC) based representation learning function described in Equation 3 minimizes the pairwise similarity between feature vectors between a positive set $A_k$ and the remaining negative sets in $\mathcal{V} \setminus A_k$ while maximizing the similarity between features in each set $A_k$. This objective bears the closest similarity to existing contrastive learners that adopt pairwise similarity metrics (refer Section 3.2.2) and jointly models inter-cluster separation and intra-cluster compactness which are effective in

overcoming class-imbalance in representation learning. Specifically, the Orthagonal Projection Loss (OPL) and a version of Triplet Loss are special cases of of the GC based loss function.

$$L_{S_f}(\theta, A_k) = \sum_{i \in A_i} \sum_{j \in \mathcal{V} \setminus A_k} S_{ij}(\theta) - \lambda \sum_{i,j \in A_k} S_{ij}(\theta), \;\; L_{C_f}(\theta, A_k) = \lambda \sum_{i \in A_k} \sum_{j \in \mathcal{V} \setminus A_k} S_{ij}(\theta) \quad (3)$$

**Log-Determinant** (LogDet) function measures the volume of a set $A_k$ in the feature space. Minimizing the LogDet over a set of datapoints in set $A_k$ shrinks the feature volume forming a tighter cluster. On the other hand, maximizing the LogDet over the entire ground-set $\mathcal{V}$ ensures diversity in the feature space which results in well separated feature clusters. The total correlation formulation $C_f$ of LogDet demonstrates the aforementioned properties which we propose as a loss function $L(\theta, A_k)$ as shown in Equation 4. The proposed objective minimizes the LogDet over the samples in a class $A_k$ while maximizing the diversity in the feature space $\mathcal{V}$.

$$L_{S_f}(\theta, A_k) = \log \det(S_{A_k}(\theta) + \lambda \mathbb{I}_{|A_k|}), \;\; L_{C_f}(\theta, A_k) = L_{S_f}(\theta, A_k) - \log \det(S_{\mathcal{V}}(\theta) + \lambda \mathbb{I}_{|\mathcal{V}|})$$
$$(4)$$

The $S_f$ version captures intra-cluster similarity, while the $C_f$ version (which empirically we see works better) captures both intra-cluster similarity and inter-cluster dissimilarity.

Adopting set-based information-theoretic functions $L(\theta)$ defined in SCoRe has been shown to outperform existing pairwise similarity-based loss functions. Through our experiments in Section 4 – we demonstrate that minimizing $L(\theta)$ for the functions discussed above is effective in forming compact and disjoint feature clusters in embedding space even under extreme class-imbalance scenarios. This proves the effectiveness of submodular functions in representation learning. We also show in the next section, that existing approaches in metric learning, are special cases of SCoRe.

### 3.2.2 SCoRe Generalizes Existing Metric/Contrastive Learning Loss Functions

In Section 2 we explore various existing approaches in metric learning and contrastive learning Khosla et al. (2020); Chen et al. (2020a) in the supervised setting. These contrastive learners mostly adopt pairwise similarity $S$ or distance $D$ metrices to learn discriminative feature sets. Interestingly, many of these existing loss functions are either special cases of SCoRe (i.e., there exist submodular functions such that the loss functions are SCoRe instantiated with those submodular functions) or closely related. We study Triplet-Loss Schroff et al. (2015), N-pairs loss Sohn (2016), SupCon Khosla et al. (2020), Orthogonal Projection Loss (OPL) Ranasinghe et al. (2021), and Soft Nearest Neighbor (SNN) Frosst et al. (2019) losses. From Table 1 we see that most contrastive learning objectives are either submodular or can be re-formulated as a submodular function. OPL and N-pairs loss are submodular, while SupCon, SNN, and Triplet losses are not naturally submodular. However, as we see in rows 7 through 9 of Table 1, we can modify these loss functions a little and we get submodular versions. We call them *Submod-Triplet* loss, *Submod-SNN* loss and *Submod-SupCon* loss. The proofs of submodularity for the proposed objectives are included in Section A.5 of the appendix. The experiments conducted on these functions in Section 4 show that the submodular variants of these objectives are better than the non-submodular counterparts, thus demonstrating the value of SCoRe.

## 4 Experiments

We perform experiments on three major settings as discussed in Section 4.1 to evaluate the effectiveness of the proposed combinatorial objectives and demonstrate their results in Sections 4.2 and 4.3.

### 4.1 Experimental Setup

**Class-Imbalanced Image Classification** : We perform our experiments on two imbalanced settings of the CIFAR-10 Krizhevsky (2009) and two naturally imbalanced subsets of MedMNIST Yang et al. (2023) datasets respectively. The results are discussed in Section 4.2. For the **CIFAR-10** we follow

---

[3]Abbreviations are included in section A.1 of the appendix.

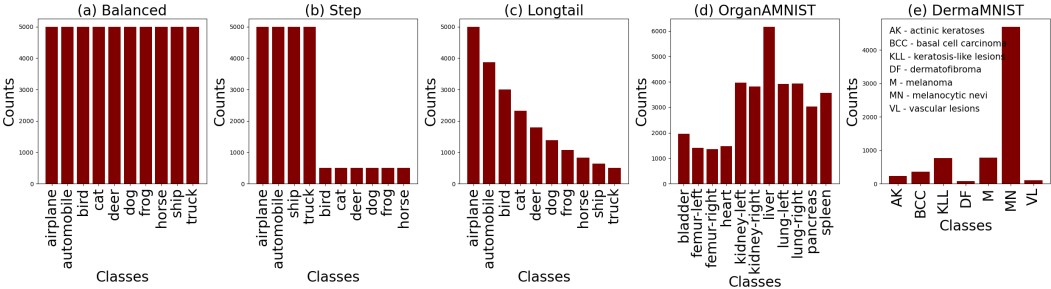

Figure 3: **Data Distribution of CIFAR-10 and MedMNIST datasets** under the class-imbalanced setting. Subfigure (a) depicts the balanced setting, (b,c) depicts the pathological class imbalanced settings in CIFAR-10 and (d,e) depict the naturally imbalanced OrganAMNIST and DermaMNIST[3]datasets respectively.

Cao et al. (2016) to create a pathological **longtail** distribution by sampling random samples using an exponentially decaying function. We also exploit the hierarchy already available in the dataset to create a **step** based imbalanced distribution. In contrast to pathological imbalance **MedMNIST** dataset demonstrates natural imbalance. We conduct our experiments on the *OrganAMNIST* and *DermaMNIST* subsets of MedMNIST due to the presence of extreme imbalance in data distributions. The data distributions of the adopted benchmarks are depicted in Figure 3. The *OrganAMNIST* dataset consists of 1-channel $[28 \times 28]$ dimensional axial slices from CT volumes, highlighting 11 distinct organ structures for a multi-class classification task. *DermaMNIST* presents dermatoscopic images of pigmented skin lesions, with 7 distinct dermatological conditions. Our proposed framework adopts the architecture and training strategy similar to Khosla et al. (2020). The backbone for the feature extractor is chosen to be a ResNet-50 He et al. (2016). The stage 1 training proceeds with training the backbone on normalized 128 dimensional feature vectors using $L(\theta)$. In stage 2 we freeze the backbone and use the output of the final pooling layer to train a linear classifier $Clf$. For every objective function we report the top-1 classification accuracy after completing two stages of model training.

Table 2: **Multi-class classification performance (Top1-Accuracy %)of submodular combinatorial objectives** (shaded in Green ) against existing approaches in metric learning and their submodular variants (shaded in blue ) on **Class-Imbalanced** *CIFAR-10* (columns 2 - 3)and *MedMNIST* (columns 4 - 5) datasets. We also compare the performance of these approaches in both the balanced and imbalanced settings.

| Objective Function | CIFAR-10 | | MedMNIST | |
| --- | --- | --- | --- | --- |
| | Pathological LongTail | Pathological Step | OrganMNIST (Axial) | DermaMNIST |
| Cross-Entropy (CE) | 86.44 | 74.49 | 81.80 | 71.32 |
| Triplet Loss Schroff et al. (2015) | 85.94 | 74.23 | 81.10 | 70.92 |
| N-Pairs Sohn (2016) | 89.70 | 73.10 | 84.84 | 71.82 |
| Lifted Structure Loss Song et al. (2016) | 82.86 | 73.98 | 84.55 | 71.62 |
| SNN Frosst et al. (2019) | 83.65 | 75.97 | 83.85 | 71.87 |
| Multi-Similarity Loss Wang et al. (2019) | 82.40 | 76.72 | 85.50 | 71.02 |
| SupCon Khosla et al. (2020) | 89.96 | 78.10 | 87.35 | 72.12 |
| Submod-Triplet (ours) | 89.20 | 74.36 | 86.03 | 72.35 |
| Submod-SNN (ours) | 89.28 | 78.76 | 86.21 | 71.77 |
| Submod-SupCon (ours) | 90.81 | 81.31 | 87.48 | 72.51 |
| Graph-Cut $[S_f]$ (ours) | 89.20 | 76.89 | 86.28 | 69.10 |
| Graph-Cut $[C_f]$ (ours) | 90.83 | 87.37 | **87.57** | 72.82 |
| LogDet $[C_f]$ (ours) | 90.80 | 87.00 | 87.00 | 72.04 |
| FL $[C_f/ S_f]$ (ours) | **91.80** | **87.49** | 87.22 | **73.77** |

**Class-Imbalanced Object Detection** - We also perform experiments on real-world class-imbalanced settings demonstrated by the India Driving Dataset Varma et al. (2019) (IDD) and LVIS (Gupta et al., 2019) in Section 4.3. IDD-Detection dataset demonstrates an unconstrained driving environment, characterized by natural class-imbalance, high traffic density and large variability among object classes. On the other hand the LVIS dataset encapsulates 1203 commonplace objects from the MS-COCO Lin et al. (2014) detection dataset with extreme imbalance among classes. We adopt the

detectron[4] framework for training and evaluating the object detection model. The architecture of the object detector is a Faster-RCNN Ren et al. (2015) model with a ResNet-101 backbone. We adopt the Feature Pyramidal Network (FPN) as in (Lin et al., 2017) to handle varying object sizes in traffic environments.

Our models are trained on 2 NVIDIA A6000 GPUs with additional details provided in Section A.2 due to space constraints. Alongside experiments on benchmark datasets we conduct experiments on a synthetic dataset described in Section A.3 of the appendix section, which demonstrate the characteristics of the newly introduced loss functions in SCoRe.

Table 3: **Object detection performance on IDD and LVIS datasets**. Applying our combinatorial objectives on a Faster-RCNN + FPN model produces the best Mean Average Precision ($mAP$) on real-world class-imbalanced settings.

| Method | Backbone and head | $mAP$ | $mAP_{50}$ | $mAP_{75}$ |
|---|---|---|---|---|
| **INDIA DRIVING DATASET (IDD)** | | | | |
| YOLO -V3[5](Redmon & Farhadi, 2018) | Darknet-53 | 11.7 | 26.7 | 8.9 |
| Poly-YOLO[4] (Hurtík et al., 2020) | SE-Darknet-53 | 15.2 | 30.4 | 13.7 |
| Mask-RCNN[4] (He et al., 2017) | ResNet-50 | 17.5 | 30.0 | 17.7 |
| Retina-Net (Lin et al., 2017) | ResNet-50 + FPN | 22.1 | 35.7 | 23.0 |
| Faster-RCNN (Ren et al., 2015) | ResNet-101 | 27.7 | 45.4 | 28.2 |
| Faster-RCNN + FPN | ResNet-101 + FPN | 30.4 | 51.5 | 29.7 |
| Faster-RCNN + SupCon | ResNet-101 + FPN | 31.2 | 53.4 | 30.5 |
| Faster-RCNN + Graph-Cut $[C_f]$ | ResNet-101 + FPN | 33.6 | 56.0 | 34.6 |
| Faster-RCNN + Facility-Location $[S_f/C_f]$ | ResNet-101 + FPN | **36.3** | **59.5** | **37.1** |
| **LVIS DATASET** | | | | |
| Faster-RCNN + FPN | ResNet-101 + FPN | 14.2 | 24.4 | 14.9 |
| Faster-RCNN + SupCon | ResNet-101 + FPN | 14.4 | 26.3 | 14.3 |
| Faster-RCNN + Graph-Cut $[C_f]$ | ResNet-101 + FPN | 17.7 | 29.1 | 18.3 |
| Faster-RCNN + Facility-Location $[S_f/C_f]$ | ResNet-101 + FPN | **19.1** | **30.5** | **20.3** |

### 4.2 RESULTS ON CLASS-IMBALANCED IMAGE CLASSIFICATION TASK

In this section , we discuss the results of training the proposed architecture in Section 4.1 on pathologically imbalanced CIFAR-10 and naturally imbalanced MedMNIST benchmarks.

**CIFAR-10** : This benchmark consists of two settings - *Longtail* and *step* as described in Section 4.1. For both the imbalanced settings in CIFAR-10, we show that submodular combinatorial objective functions outperform SoTA metric learners like SupCon by upto 2% (shown by FL) for the *longtail* distribution and upto 7.6% (shown by FL) for the *step* distribution. Amongst existing contrastive learning approaches, objective functions which consider multiple positive and negative pairs (SupCon) demonstrate significant performance improvements. The reformulated submodular objectives - *Submod-Triplet*, *Submod-SNN* and *Submod-SupCon* demonstrate upto 3.5%, 3.7% and 4.11% respectively over their non-submodular counterparts.

**MedMNIST** : The natural class imbalance and the large variability in patient data demonstrated in MedMNIST serves as a playground for highlighting the effectiveness of our combinatorial objectives in overcoming class imbalance. As discussed in Section 4.1 we consider two popular subsets of the MedMNIST dataset - *DermaMNIST* and *OrganAMNIST*, the data-distribution for which has been depicted in Figure 3 (d,e). For both *OrganAMNIST* and *DermaMNIST* subsets discussed in Section 4.1, we show that proposed combinatorial objectives in SCoRe outperform SoTA approaches by upto 0.25% (as shown in GC) and 1.5% ( as shown in FL) respectively. For the OrganAMNIST dataset, the lack of variability in features (2D images) between classes leads to a smaller gain in performance, maximum being 0.25% by the Graph-Cut based objective function. Similar to CIFAR-10 benchmark we also observe that, reformulated submodular objectives of existing contrastive losses consistently outperform their non-submodular counterparts.

### 4.3 RESULTS ON OBJECT DETECTION TASK ON REAL-WORLD CLASS-IMBALANCED SETTING

We benchmark the performance of our approach against SoTA object detectors which adopt Focal Loss Lin et al. (2017), data-augmentations etc. At first, we introduce a contrastive learning based objective (SupCon) in the box classification head of the object detector and show that contrastive

---

[4]https://github.com/facebookresearch/detectron2
[5]Results are from Hurtík et al. (2020).

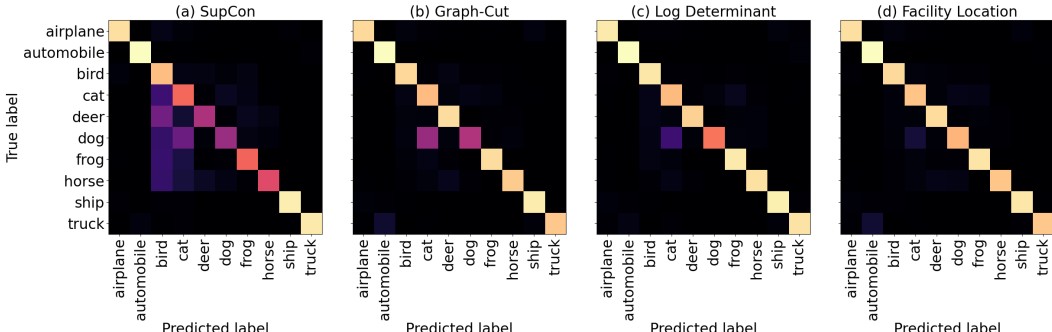

Figure 4: **Comparison of Confusion Matrix plots** between (a) SupCon Khosla et al. (2020), (b) Graph-Cut (GC), (c) Log Determinant, and (d) Facility Location (FL) for the longtail imbalanced setting of CIFAR-10 dataset. We show significant reduction in inter-class bias when employing combinatorial objectives in SCoRe characterized by reduced confusion between classes.

learning outperforms standard model training (using CE loss) on IDD by 3.6 % (1.9 $mAP_{50}$ points) and 10.6% (2.8 $mAP_{50}$ points) on LVIS datasets. Secondly, we introduce the objectives in SCoRe to the box classification head and show that they outperform the SoTA as well as the contrastive learning objective Khosla et al. (2020). The results in Table 3 show that the Facility Location and Graph-Cut based objective function outperforms the SoTA method by 6.1 $mAP_{50}$ and 2.6 $mAP_{50}$ points respectively on IDD, alongside 6.1 $mAP_{50}$ and 4.7 $mAP_{50}$ points respectively on LVIS. Additionally, from the class-wise performance on IDD as shown in Figure 1, submodular combinatorial objectives demonstrate a sharp rise in performance ($mAP_{50}$ value) of the rare classes (a maximum of 6.4 $mAP$ points for *Bicycle* class) over the contrastive objective.

### 4.4 DOES COMBINATORIAL LOSS FUNCTIONS FORM BETTER CLUSTERS ?

As discussed in Section 3, real-world setting like MedMNIST and IDD, introduce inter-class bias and intra-class variance during model training as a resultant of class-imbalance. Confusion matrix plots on predicted class labels after stage 2 of model training are used to compare the objective functions studied in SCoRe to form disjoint clusters, therby overcoming class-imbalance. We compare between SoTA approach SupCon, and proposed GC, LogDet and FL based objective functions for the longtail imbalanced CIFAR-10 dataset. Plots in Figure 4 show that SupCon shows 2̃2% overall confusion with elevated confusion between the *animal* hierarchy of CIFAR-10. A significant drop in confusion is observed in combinatorial objectives with a minimum of 8.2% for FL. Both GC and LogDet demonstrate confusion between structurally similar objects like *cat* and *dog* (4-legged animals). As discussed in Majee et al. (2021), the reduction in confusion by objectives proposed in SCoRe shows a reduction in inter-class bias. This is correlated to reducing the impact of class-imbalance due to formation of discriminative feature clusters. Thus we show that Submodular combinatorial objectives are a better choice over SoTA metric learners for representation learning tasks.

## 5 CONCLUSION

We introduce a family of submodular combinatorial objectives for representation learning tasks through the SCoRe framework to overcome class imbalance in real-world vision tasks. The proposed combinatorial objectives drive a paradigm shift in training strategy from pairwise distance or similarity matrices in SoTA approaches to set-based loss functions in SCoRe. The proposed SCoRe frameowork also highlights that existing approaches in metric/contrastive learning are either submodular in nature or can be reformulated into submodular forms. We conduct our experiments on two image classification benchmarks, namely, pathologically imbalanced CIFAR-10 and naturally imbalanced MedMNIST, alongside two real-world unconstrained object detection benchmark, namely the Indian Driving Dataset (IDD) and LVIS. Our proposed combinatorial loss functions outperform existing SoTA approaches on all three benchmarks by upto 7.6% for the classification task and 19.4% on the detection task. Our experiments also suggest that combinatorial counterparts of existing objectives outperform their original functions by significant margins. This establishes the importance of combinatorial loss functions in overcoming class-imbalance and its underlying pitfalls in representation learning tasks.

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

# A APPENDIX

## A.1 NOTATIONS

Following the problem definition in Section 3.1 we introduce several important notations in Table 4 that are used throughout the paper.

## A.2 EXPERIMENTAL SETUP : ADDITIONAL INFORMATION

**Class-Imbalanced Image Classification** : The SCoRe framework introduces two pathological imbalance settings - *Longtail* and *Step* in the CIFAR-10 benchmark. We create the **longtail distribution** by sampling random samples using an exponentially decaying function. The decay rate of the function is set to $1/10$ which results in the longtail subclass to have 600 samples while the abundant subclass has 6000 samples. The **step** function based imbalance setting exploits the hierarchy already available in the dataset. The CIFAR-10 dataset can be broadly classified into *animal* and *automobile* classes. We use this information to subsample the *animal* (chosen at random) class objects to create an imbalanced **step** data distribution. The distributions of the dataset is depicted in Figure 3. We train our models in stage 1 with a batch size of 512 (1024 after augmentations) with an initial learning rate of 0.4, trained for 1000 epochs with a cosine annealing scheduler. In stage 2 we freeze the backbone and use the output of the final pooling layer to train a linear classifier $Clf$ with a batch size of 512 and a constant learning rate of 0.8.

**Class-Imbalanced Medical Image Classification** : In contrast to pathological imbalance introduced in CIFAR-10 we benchmark our proposed objectives in the SCoRe framework against SoTA approaches in contrastive learning on two subsets of MedMNIST Yang et al. (2023) dataset. The *OrganAMNIST* dataset consists of axial slices from CT volumes, highlighting 11 distinct organ structures for a multi-class classification task. Each image is of size $[1 \times 28 \times 28]$ pixels. The *DermaMNIST* subset presents dermatoscopic images of pigmented skin lesions, also resized to $[3 \times 28 \times 28]$ pixels.

Table 4: Collection of notations used in the paper.

| Symbol | Description |
|---|---|
| $\mathcal{T}$ | The training Set. $|\mathcal{T}|$ denotes the size of the training set. |
| $\mathcal{V}$ | Ground set containing feature vectors from all classes in $\mathcal{T}$. |
| $F(x, \theta)$ | Convolutional Neural Network used as feature extractor. |
| $Clf(., .)$ | Multi-Layer Perceptron as classifier. |
| $\theta$ | Parameters of the feature extractor. |
| $S_{ij}(\theta)$ | Similarity between images $i, j \in \mathcal{T}$. |
| $D_{ij}(\theta)$ | Distance between images $i, j \in \mathcal{T}$. |
| $p$ | Positive sample which is of the same class $c_i$ as the anchor $a$. |
| $n$ | Negative sample which is of the same class $c_i$ as the anchor $x$. |
| $A_k$ | Target set containing feature representation from a single class $k \in c_i$. |
| $f(S_{ij})$ | Submodular Information function over the similarity kernel $S$. |
| $S_f$ | Variant of submodular information function denoting total information in the ground set $V$. |
| $C_f$ | Variant of submodular information function denoting total correlation in the ground set $V$. |
| $L(\theta)$ | Loss value computed over all classes $c_i \in C$. |
| $L(\theta, A_k)$ | Loss value for a particular set/class $A_k$ given parameters $\theta$. |
| AK | Actinic Keratoses |
| BCC | Basal Cell Carcinoma |
| KLL | Keratosis-Like-Lesions |
| DF | Dermatofibroma |
| M | Melanoma |
| MN | Melanocytic Nevi |
| VL | Vascular Lesions |

This dataset supports a multi-class classification task with 7 different dermatological conditions. The OrganAMNIST dataset contains 34581 training and 6491 validation samples of single channel images highlighting various modalities of 8 different organs. Although the DermaMNIST has RGB images, it is a small scale dataset with a total of 7007 training samples and 1003 validation samples. For both these subsets used in our framework, pixel values were normalized to the range $[0, 1]$, and we relied on the standard train-test splits provided with the datasets for our evaluations. The results from the experiments are discussed in Section 4.2.

**Class-Imbalanced Object Detection** - IDD-Detection dataset demonstrates an unconstrained driving environment, characterized by natural class-imbalance, high traffic density and large variability among object classes. This results in the presence of rare classes like *autorickshaw*, *bicycle* etc. and small sized objects like *traffic light*, *traffic sign* etc. There are a total of 31k training images in IDD and 10k validation images of size $[3 \times 1920 \times 1080]$ with high traffic density, occlusions and varying road conditions. The architecture of the object detector is a Faster-RCNN Ren et al. (2015) model with a ResNet-101 backbone alongside the Feature Pyramidal Network (FPN) as in (Lin et al., 2017) to handle varying object sizes in traffic environments. Our framework also draws inspiration from FSCE **?** with proposed modifications to existing Faster-RCNN + FPN based detectors. During the fine-tuning process on Imbalanced datasets we keep the Region Proposal Network (RPN) and the ROI pooling layers unfrozen to adapt to the rare classes. We also double the maximum number of proposals kept after Non-Maximal Suppression (NMS), bringing more proposals from rare classes to the foreground. We consider only half the number of proposals from the ROI pooling layer (top 256 out of 512) for computing the loss function. This forces the objective function to better penalize the object detector for predicting low objectness scores for objects belonging to the rare classes. The model is trained for 17000 iterations with a batch size of 8 and an initial learning rate of 0.02. A step based learning rate scheduler is adopted to reduce the learning rate by 1/10 at regular intervals.

Similar to IDD , the LVIS Gupta et al. (2019) dataset depicts an extreme case of longtail imbalance with a large number of tail classes. The dataset consists of 1203 classes created by extending the label set in MS-COCO Lin et al. (2014) (consisting of just 80 classes). We adopt the version v1.0 of LVIS for our experiments and conduct our experiments on Faster-RCNN with a batch size of 16, initial learning rate of 0.06 with repeat factor sampling for a total of 180,000 iterations. Similar to IDD , the step based learning rate scheduler is adopted to reduce the learning rate by 1/10 at regular intervals.

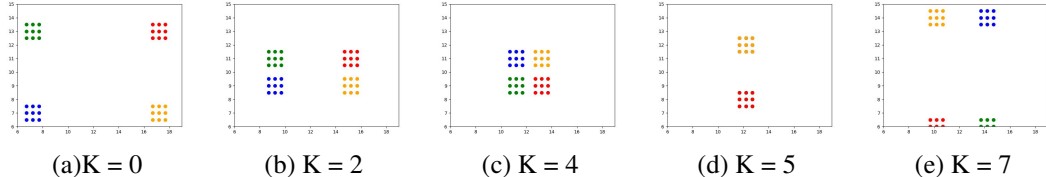

(a)K = 0      (b) K = 2      (c) K = 4      (d) K = 5      (e) K = 7

Figure 5: **Plot of the data distribution of the synthetic dataset with respect to varying values of** $K$. The synthetic dataset has four clusters. The value of $K$ determines the distance between the cluster centroids. Between values 0 and 4 the cluster separation reduces and between 5 and 7 the inter-cluster separation increases.

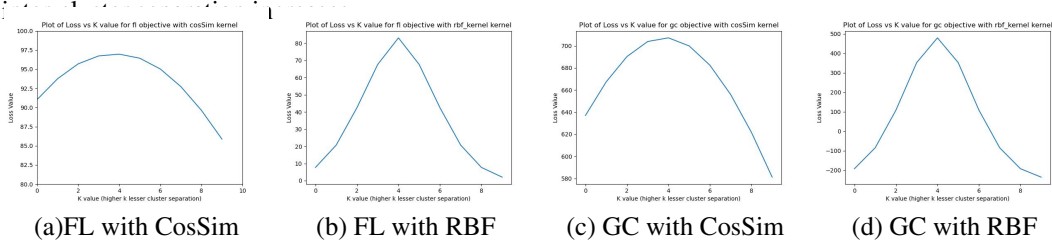

(a)FL with CosSim      (b) FL with RBF      (c) GC with CosSim      (d) GC with RBF

Figure 6: **Variation in the Value of the combinatorial loss $L(\theta)$ against different values of $K$** with different similarity kernels, (a) FL objective with Cosine Similarity (CosSim) kernel, (b) FL objective with Radial Bias Field (RBF) kernel, (c) GC objective with CosSim kernel, and (d) GC objective with RBF kernel. We observe an increase in loss value with reduction in inter-cluster separation and vice-versa.

## A.3 Experiments on Synthetic a dataset

This experiment characterizes the proposed submodular combinatorial objectives by demonstrating the variation in the loss value under varying inter-cluster separations and similarity kernels. The experiment proceeds with the creation of four orthogonal clusters projected on a 2-dimensional feature space with sufficient inter-cluster separation (denoted by K = 0) as shown in Figure 5(a). Over successive rounds we reduce the inter-cluster separation (by increasing the value of K from 0 through 5) such that overlaps exist between feature clusters from Figures 5(a) through (c). Further we increase the inter-cluster separation in the opposite direction (by increasing the value of K beyond 4) as shown in Figures 5(d) and (e).

For a chosen similarity kernel (either cosine-similairty or RBF Killamsetty et al. (2021)) we plot the calculated loss values $L(\theta)$ as in Figure 5 under varying cluster separations as discussed above. We observe that as the inter-cluster separation reduces, the value of $L(\theta)$ increases and vice-versa. This holds true irrespective of the choice of similarity kernels. Thus, by minimizing the combinatorial objective would result in large-inter feature clusters establishing the efficacy of our proposed combinatorial objectives.

Table 5: **Ablation study** on the effect of $\lambda$ on the performance of Graph-Cut based combinatorial objective in SCoRe.

| $\lambda$ | Top-1 acc CIFAR-10 (longtail) |
|---|---|
| 0.5 | 83.65 |
| **1.0** | **89.96** |
| 1.5 | 87.11 |
| 2.0 | 85.86 |

## A.4 Ablation Study: Effect of $\lambda$ on performance of Graph-Cut based Objective

In this section we perform experiments on the hyperparameter $\lambda$ introduced in Graph-Cut based combinatorial objective in SCoRe. The hyper-parameter $\lambda$ is applied to the sum over the penalty associated with the positive set forming tighter clusters. This parameter controls the degree of compactness of the feature cluster ensuring sufficient diversity is maintained in the feature space. For GC to be submodular it is also important for $\lambda$ to be greater than or equal to 1 ($\lambda \geq 1$). For this experiment we train the two stage framework in SCoRe on the logtail CIFAR-10 dataset for 500

epochs in stage 1 with varying $\lambda$ values in range of [0.5, 2.0] and report the top-1 accuracy after stage 2 model training on the validation set of CIFAR-10. Table 5 shows that we achieve highest performance for $\lambda = 1$ for longtail image classification task on the CIFAR-10 dataset. We adopt this value for all experiments conducted on GC in this paper.

## A.5 PROOF OF SUBMODULARITY

In this section we discuss in depth the submodular counterparts of three existing objective functions in contrastive learning. We provide proofs that these functions are non-submodular in their existing forms and can be reformulated as submodular objectives through modifications without changing the characteristics of the loss function.

### A.5.1 TRIPLET LOSS AND SUBMOD-TRIPLET LOSS

**Triplet Loss** : We first show that the Triplet loss is not necessarily submodular. The reason for this is the Triplet loss is of the form: $\sum_{i,p\in A, n\in \mathcal{V}\setminus A} D_{ipn} = \sum_{n\in \mathcal{V}} \sum_{i,p\in A} D_{ipn} - \sum_{i,p,n\in A} D_{ipn}$. Note that this is actually supermodular since $-\sum_{i,p\in A} D_{ipn}$ is submodular and $\sum_{i,p,n\in A} D_{ipn}$ is submodular. As a result, the Triplet loss is **not necessarily submodular**.

**Submod-Triplet** : Submodular Triplet loss (Submod-Triplet) is exactly the same as Graph-Cut where we use $\lambda = 1$ and the similarity as the squared similarity function. Thus, this function is **submodular** in nature.

### A.5.2 SOFT-NEAREST NEIGHBOR (SNN) LOSS AND SUBMOD-SNN LOSS

**SNN Loss** : From the set representation of the SNN loss we can describe the objective $L(\theta, A_k)$ as in Equation 5 . This objective function can be split into two distinct terms labelled as *Term 1* and *Term 2* in the equation above.

$$L(\theta, A_k) = -\sum_{i\in A_k} [\underbrace{\log \sum_{j\in A_k} \exp(S_{ij}(\theta))}_{\text{Term 1}} - \underbrace{\log \sum_{j\in \mathcal{V}\setminus A_k} \exp(S_{ij}(\theta))}_{\text{Term 2}}] \tag{5}$$

We prove the objective to be submodular by considering two popular assumptions :
(1) The sum of submodular function over a set of classes $A_i$, $i \in C$, the resultant is submodular in nature.
(2) The concave over a modular function is submodular in nature.
To prove that $L(\theta, A_k)$ is submodular in nature it is enough to show the individual terms (Term 1 and 2) to be submodular. Note that the sum of submodular functions is submodular in nature. Considering $F(A) = \sum_{j\in A_k} S_j$ for any given $i \in A_k$, we see that $\log \sum_{j\in A_k} \exp(D_j(\theta))$ to be modular as it is a sum over terms $\exp(D_j(\theta))$.
We also know from assumption (2) Fujishige (2005a), that the concave over a modular function is *submodular* in nature, log being a concave function. Thus, $log \sum_{j\in A_k} exp(S_j)$ is submodular function for a given $i \in A_k$. Unfortunately, the negative sum over a submodular function cannot be guaranteed to be submodular in nature. This renders SNN to be **non-submodular** in nature.

**Submod-SNN Loss** : The variation of SNN loss described in Table 1 can be represented as $L(\theta, A_k)$ as shown in Equation 6. Similar to the set notation of SNN loss we can split the equation into two terms, referred to as *Term 1* and *Term 2* in the equation above.

$$L(\theta, A_k) = \sum_{i\in A_k} [\underbrace{\log \sum_{j\in A_k} \exp(D_{ij}(\theta))}_{\text{Term 1}} + \underbrace{\log \sum_{j\in \mathcal{V}\setminus A_k} \exp(S_{ij}(\theta))}_{\text{Term 2}}] \tag{6}$$

Considering $F(A) = \sum_{j\in A_k} S_j$ for any given $i \in A_k$, we prove $\log \sum_{j\in A_k} \exp(D_j(\theta))$ to be modular, similar to the case of SNN loss. Further, using assumption (2) mentioned above we prove that the log (a concave function) over a modular function is submodular in nature. Finally, the sum of submodular functions over a set of classes $A_k$ is submodular according to assumption (1). Thus the term 1, $\sum_{i\in A_k} \log \sum_{j\in A_k} \exp(D_{ij}(\theta))$ in the equation of Submod-SNN is proved to be submodular in nature.

The term 2 of the equation represents the total correlation function of Graph-Cut ($L_{C_f}(\theta, A_k)$). Since graph-cut function has already been proven to be submodular in (Fujishige, 2005a; Iyer et al., 2022) we prove that term 2 is submodular.

Finally, since the sum of submodular functions is submodular in nature, the sum over term 1 and term 2 which constitutes $L(\theta, A_k)$ can also be proved to be **submodular**.

### A.5.3 N-Pairs Loss and Orthogonal Projection Loss (OPL)

In Table 1 both N-pairs loss and OPL has been identified to be submodular in nature. In this section we provide proofs to show they are submodular in nature.

**N-pairs Loss** : The N-pairs loss $L(\theta, A_k)$ can be represented in set notation as described in Equation 7. Similar to SNN loss, we can split the equation into two distinct terms.

$$L(\theta, A_k) = -[\underbrace{\sum_{i,j \in A_k} S_{ij}(\theta)}_{\text{Term 1}} + \underbrace{\sum_{i \in A_k} log(\sum_{j \in \mathcal{V}} S_{ij}(\theta) - 1)}_{\text{Term 2}}] \tag{7}$$

The first term (Term 1) in N-pairs is a negative sum over similarities, which is submodular in nature Fujishige (2005a). The second term (Term 2) is a log over $\sum_{j \in \mathcal{V}} S_{ij}(\theta) - 1$, which is a constant term for every training iteration as it encompases the whole ground set $\mathcal{V}$. The sum of Term 1 and Term 2 over a set $A_k$ is thus **submodular** in nature.

**OPL** : The loss can be represented as Equation 8 in its original form. Similar to above objectives we split the equation into two distinct terms and individually prove them to be submodular in nature.

$$L(\theta, A_k) = (1 - \underbrace{\sum_{i,j \in A_k} S_{ij}(\theta)}_{\text{Term 1}}) + (\underbrace{\sum_{i \in A_k} \sum_{j \in \mathcal{V} \setminus A_k} S_{ij}(\theta)}_{\text{Term 2}}) \tag{8}$$

The Term 1 represents a negative sum over similarities in set $A_k$ and is thus submodular in nature. The Term 2 is exactly $L_{C_f}$ of Graph-Cut (GC) with $\lambda = 1$ and is also submodular in nature. Since the sum of two submodular functions is also submodular, $L(\theta, A_k)$ in Equation 8 is also **submodular**.

### A.5.4 SupCon and Submod-SupCon

**SupCon** : The combinatorial formulation of SupCon as in Equation 9 can be defined as a sum over the set-function $L(\theta, A_k)$ as described in Table 1 of the main paper.

$$L(\theta, A_k) = \underbrace{\frac{-1}{|A_k|} \sum_{i,j \in A_k} S_{ij}}_{\text{Term 1}} + \underbrace{\sum_{i \in A_k} \frac{1}{|A_k|} log(\sum_{j \in V} exp(S_{ij}) - 1)}_{\text{Term 2}} \tag{9}$$

Similar to earlier objectives SupCon can be split into two additive terms and it is deemed enough to show that individual terms in the equations are submodular in nature. Let the marginal gain in set $A$ on addition of new element x be denoted as $F(\theta, x|A)$ where $x \in V \setminus A$. $F(\theta, x|A)$ can be written as in Fujishige (2005a) as $F(\theta, A \cup \{x\}) - F(\theta, A)$. To prove that $F(\theta, A_i)$ is submodular it is enough to prove the condition of diminishing marginal returns, $F(\theta, x|A) \geq F(\theta, x|B)$, where $A \subseteq B \subseteq V$ and $x \in V \setminus B$. Now, considering Equation 9 we can compute the marginal gain when $\{x\}$ is added to $A$ as follows:

$$F(\theta, x|A) = F(\theta, A \cup \{x\}) - F(\theta, A)$$

$$= \frac{-1}{|A \cup \{x\}|} \sum_{i,j \in A \cup \{x\}} S_{ij} + \sum_{i \in A \cup \{x\}} \frac{1}{|A \cup \{x\}|} log\left(\sum_{j \in V} exp(S_{ij}) - 1\right) - F(\theta, A)$$

$$= \left[\frac{-1}{|A \cup \{x\}|} \sum_{i,j \in A \cup \{x\}} S_{ij}\right] - \left[\frac{-1}{|A|} \sum_{i,j \in A} S_{ij}\right]$$

$$+ \left[\sum_{i \in A \cup \{x\}} \frac{1}{|A \cup \{x\}|} log\left(\sum_{j \in V} exp(S_{ij}) - 1\right)\right] - \left[\sum_{i \in A} \frac{1}{|A|} log\left(\sum_{j \in V} exp(S_{ij}) - 1\right)\right]$$

To prove the first term (Term 1) to be submodular we need to show that the marginal gain on adding $\{x\}$ to set $A$ is greater than or equal to that in $B$. This can be expressed as the below inequality:

$$\left[\frac{-1}{|A \cup \{x\}|} \sum_{i,j \in A \cup \{x\}} S_{ij}\right] - \left[\frac{-1}{|A|} \sum_{i,j \in A} S_{ij}\right] \geq \left[\frac{-1}{|B \cup \{x\}|} \sum_{i,j \in B \cup \{x\}} S_{ij}\right] - \left[\frac{-1}{|B|} \sum_{i,j \in B} S_{ij}\right] \tag{10}$$

Simplifying the Left-Hand-Side of the equation we get:

$$= \left[\frac{-1}{|A \cup \{x\}|} \sum_{i,j \in A \cup \{x\}} S_{ij}\right] - \left[\frac{-1}{|A|} \sum_{i,j \in A} S_{ij}\right]$$

$$= \left[\frac{-1}{|A|+1} \sum_{i,j \in A} S_{ij}\right] + \left[\frac{-1}{|A|+1} \sum_{i \in A} S_{ix}\right] + \left[\frac{-1}{|A|+1} \sum_{j \in A} S_{xj}\right] - \left[\frac{-1}{|A|} \sum_{i,j \in A} S_{ij}\right]$$

$$= \left[\frac{-2}{|A|+1} \sum_{i \in A} S_{ix}\right] + \left[\frac{1}{|A|(|A|+1)} \sum_{i,j \in A} S_{ij}\right] + \left[\frac{-1}{|A|+1}\right]$$

Substituting the above equation in 10 we get:

$$\left[\frac{-2}{|A|+1} \sum_{i \in A} S_{ix}\right] + \left[\frac{1}{|A|(|A|+1)} \sum_{i,j \in A} S_{ij}\right] + \left[\frac{-1}{|A|+1}\right]$$

$$\geq \left[\frac{-2}{|B|+1} \sum_{i \in B} S_{ix}\right] + \left[\frac{1}{|B|(|B|+1)} \sum_{i,j \in B} S_{ij}\right] + \left[\frac{-1}{|B|+1}\right]$$

From the above inequality we see that as the size of $A$ and $B$ increases due to addition of elements to individual subsets, the inequality fails to hold. This is due to the normalization terms in the denominator which increases linearly with increase in size of the individual subsets. This renders this term to be **not submodular** in nature. Since, both terms in Equation 9 needs to be submodular to show $F(\theta, A_i)$ to be submodular, we can conclude the SupCon Khosla et al. (2020) is not submodular in nature.

**Submod-SupCon** : The submodular SupCon as shown in Equation 11 can be split into two terms indicated as Term 1 and Term 2.

$$L(\theta, A_k) = -[\underbrace{\sum_{i,j \in A_k} S_{ij}(\theta)}_{\text{Term 1}}] + \underbrace{\sum_{i \in A_k}[log(\sum_{j \in \mathcal{V} \backslash A_k} \exp(S_{ij}(\theta)))]}_{\text{Term 2}} \tag{11}$$

The Term 1 of Submod-SupCon is a negative sum over similarities of set $A_k$ and is thus submodular. The Term 2 of the equation is also submodular as it is a concave over the modular term $\sum_{j \in \mathcal{V} \backslash A_k} \exp(S_{ij}(\theta))$, with $\log$ being a concave function. Thus, Submod-SupCon is also **submodular** as the sum of two submodular functions is submodular in nature.

