# OpenReview forum: "SCoRe: Submodular Combinatorial Representation Learning for Real-World Class-Imbalanced Settings"
_ICLR.cc/2024/Conference — Submitted to ICLR 2024_

### Official Review · Reviewer_q4FE · 2023-10-30

**Soundness:** 2 fair
**Presentation:** 3 good
**Contribution:** 2 fair
**Rating:** 5
**Confidence:** 3

**Summary:**

This paper proposes a family of submodular combinatorial objectives for representation learning tasks through the submodular combinatorial representation learning framework to overcome class imbalance in real-world vision tasks. The authors conduct experiments on two benchmark datasets to show the effectiveness of the proposed approach.

**Strengths:**

1. This paper is well-written and easy to read.
2. The performance seems good compared with other approaches.

**Weaknesses:**

1. The novelty is unclear. The method part only lists some existing metric learning losses.
2. The proposed framework is called the Submodular Combinatorial Representation learning framework. What does Combinatorial mean? It is unclear what the framework looks like since there are only some metric learning loss functions in the method part.
3. The authors do not compare with the recent state-of-the-art method since the latest method in Table 2 is in 2020.

**Questions:**

see the weakness

---

> ### Author Response · Authors · 2023-11-22
>
> **W1. The novelty is unclear. The method part only lists some existing metric learning losses.**
>
> **A1.** We thank the reviewer for raising this concern. Our paper introduces a novel family of objective functions based on set-based submodular information measures. **The paradigm shift in machine learning to adopt set-based information functions as learning objectives and exploiting their combinatorial properties to overcome inter-class bias and intra-class variance is the key motivation of SCoRe.**
> 1. Unlike existing contrastive learning objectives which consider pairwise distance / similarity metrics to measures similarity / dissimilarity between object classes, we propose a novel paradigm in representation learning by considering each class in the training dataset $\mathcal{T}$ as a set $A_k$, where $k \in \[1,C\]$.
>
> 2. To the best of our knowledge, we are the first to introduce set-based (combinatorial) information theoretic measures as representation learning objectives - Facility Location (FL), Graph-Cut (GC) and LogDet (LogDet) which demonstrates significantly superior performance over SoTA methods.
> 3. Through theoretical proofs, we show that existing contrastive learners are either submodular in nature, or can be reformulated into submodular functions which demonstrate better performance on real-world, class-imbalanced datasets.
>
> 4. The SCoRe framework encapsulates all novel combinatorial objectives proposed in SCoRe into a generalizable framework allowing researchers in this field to explore the capabilities of combinatorial objectives on various datasets and feature extractors (backbones).
>
> We hope that the above points clarify the differences between our proposed approach and the existing metric / contrastive learners and illustrate the novelty of our proposed methodology.
>
> **W3. The authors do not compare with the recent state-of-the-art method since the latest method in Table 2 is in 2020.**
>
> **A3.** We thank the reviewer for pointing this out. Following the recommendations from reviewer zWRY we have included a literature survey of the existing approaches in longtail recognition alongside modern contrastive learners like MoCo, MoCo v2, BYOL etc. in Section 2 of the main paper. For contrastive learners, we observe a surge in modern architectures such as Vision Transformers (Dosovitskiy et al., 2021) , data augmentation based techniques and usage of auxiliary networks in recent years. Unfortunately, due to lack of compute we are unable to conduct experiments on models with such large parameter counts.
> Also our method focuses mostly on the supervised setting and to the best of our knowledge Supervised Contrastive Learner (SupCon, Khosla et al., 2020) continues to be the SoTA objective function as it continues to be adopted in SoTA architectures like GPaCo (Cui et al., 2023), GLMC (Du et al., 2023), BCL (Zu et al., 2022)etc.

---

> ### Author Response · Authors · 2023-11-22
>
> **W2. The proposed framework is called the Submodular Combinatorial Representation learning framework. What does Combinatorial mean? It is unclear what the framework looks like since there are only some metric learning loss functions in the method part.**
>
> **A2.** We thank the reviewer for raising this question. By definition, combinatorics is a branch of mathematics which encapsulates functions that deal with operations such as counting, arranging and analyzing discrete structures. In the context of our paper, combinatorial functions refer to functions in information theory that learn feature information from a set of classes in the training dataset.
>
> Objective functions introduced in SCoRe such as Facility Location (FL), Graph-Cut (GC) and Log-Determinant (LogDet) have been shown to be combinatorial functions in Fujishige, (2005). For example, applying facility location on a feature set $A_k$ which is a subset of the ground set $\mathcal{V}$ selects the most discriminative set of features (analogous to facilities in combinatorial optimization) thus uniquely identifying the class $A_k$ from classes in $\mathcal{V} \setminus A_k$.
>
> Traditionally combinatorial optimization is performed using discrete optimization techniques like greedy search, stochastic greedy search etc. (Iyer et al., 2021). By choosing the similarity metric $S_{ij}(\theta)$ to be cosine similarity as shown in Section 3.2.1 we observe that the FL function is differentiable in continuous optimization space while exhibiting the combinatorial property described above. Note, that the ‘max’ in facility location has been smoothened in our paper using the approximation adopted in Song et al., (2017), making $L(\theta, A_k) = \sum_{i \in \mathcal{V} \setminus A_k} log \sum_{j \in A_k} exp(S_{ij}(\theta))$. This allows FL based objective function to be optimized using Stochastic Gradient Descent (SGD) for training neural networks.
>
> To further demonstrate this we have included an experiment in section A.3 of the appendix by employing synthetic data and plotting the facility location information measure for varying levels of cluster overlap. The nature of the plot shows that submodular functions (FL, GC and LogDet) to be continuous and differentiable (Petersen and Pedersen, 2008). This shows the applicability of submodular functions as an objective in representation learning tasks to learn discriminative feature sets for each class (represented as $A_k$) in the training dataset $\mathcal{T}$.
> Unlike existing contrastive learners / metric learners which learn similarity / dissimilarity between image pairs, objective functions in SCoRe minimize the total information over each set (class) in the dataset enforcing intra-class compactness while maximizing the information over the ground set to enforce inter-cluster separation.
>
> The SCoRe framework provides the necessary tools to fellow researchers in this field to experiment with combinatorial objective functions and contrast their performance against existing objectives across multiple datasets and backbones. The learning framework has been adapted from Khosla et al., (2020) and has three major components:
> 1. **Feature Extractor**, $F(I, \theta)$ is a convolutional neural network which projects an input image $I$ into a $D_{f}$ dimensional feature space, $r = F(I, \theta) \in R^{D_{f}}$ given parameters $\theta$. The modular design of SCoRe allows researchers to explore multiple network architectures like ResNet, AlexNet, VGGNet etc.
> 2. **Classifier**, $Clf(r, \theta)$ is a linear projection layer that projects the $D_{f}$ dimensional input features to a smaller dimensional vector $D_{p}$, $z = Clf(r, \theta) \in R^{D_{p}}$  such that a linear classifier can classify the input image $I$ to its corresponding class label $c_{i}$ for $i \in [1, C]$.
> 3. **Combinatorial Objective Function**, $L(\theta)$ trains the feature extractor $F$ over all classes $C$ in $D$ to discriminate between classes in a multi-class classification setting. By varying the objective function we are able to study their behavior in learning discriminative feature sets for each class in $D$. SCoRe provides an unified platform to researchers to contrast existing objective functions against set-based combinatorial objectives.
>
> These details have been updated in Section 3.2 of the main paper for better clarity. Since the training scheme has been majorly adapted from Khosla et al., (2020), we include the details of the learning objective in Section A.2 of the appendix.

---

### Official Review · Reviewer_KzrN · 2023-10-30

**Soundness:** 3 good
**Presentation:** 3 good
**Contribution:** 2 fair
**Rating:** 5
**Confidence:** 3

**Summary:**

This paper focuses on improving the way deep learning models handle imbalanced class scenarios in real-world applications. In such situations, where some classes are rare, conventional neural networks struggle to learn useful features. This leads to a significant imbalance between rare and abundant classes in the data. To address this, the paper introduces the SCoRe framework, which utilizes Submodular Combinatorial Loss functions. These functions can effectively model feature diversity and cooperation among classes. Experimental results on image classification tasks, including imbalanced datasets like CIFAR-10 and object detection tasks, show that the proposed approach outperforms existing metric learning methods.

**Strengths:**

- The paper introduces a new approach to tackle the challenge of class-imbalanced data in deep learning, which is a critical problem in real-world applications.

- This paper is generally easy to follow.

**Weaknesses:**

- Unclear Link Between Diversity and Robust Representations: While the paper's motivation to employ submodular functions as loss functions to promote diversity is evident, the direct connection between diversity and the creation of robust representations from imbalanced datasets remains somewhat ambiguous. The paper does not clearly elucidate how fostering diversity contributes to the development of robust representations in such scenarios.

- Limited Experimental Evidence: The experimental results exhibit certain weaknesses:
a) The paper compares its approach with well-known metric learning methods but does not utilize popular metric learning datasets, which could potentially limit the generalizability of the findings.
b) All experiments are conducted on relatively small datasets, as opposed to widely recognized datasets commonly used in imbalanced classification, such as ImageNet-LT. This choice of datasets might limit the broader applicability and relevance of the research.

**Questions:**

See weakness.

---

> ### Author Response · Authors · 2023-11-22
>
> **W1. Unclear Link Between Diversity and Robust Representations: While the paper's motivation to employ submodular functions as loss functions to promote diversity is evident, the direct connection between diversity and the creation of robust representations from imbalanced datasets remains somewhat ambiguous. The paper does not clearly elucidate how fostering diversity contributes to the development of robust representations in such scenarios.**
>
> **A1.** We thank the reviewer for the detailed comment. We do agree that diversity among class specific features alone cannot be a metric for robustness in learnt representations.
> Our paper points out (in Section 1) that inter-cluster separation alongside intra-class compactness has to be enforced during representation learning.
> To verify this hypothesis in SCoRe, we introduce two variants of information measures as objectives based on (1) total information $L_{S_f}$ and (2) total correlation $L_{C_f}$ respectively. In summary, the $L_{S_f}$ maximizes the diversity within each object class $A_k$ by minimizing the total information in $A_k$, while the $C_f$ variant maximizes both diversity within each class and inter-class separation (by maximizing $ f(\cup_{k = 1}^{|C|} A_k)$).
> Thus $L_{C_f}$ emerges as a better variant to learn both diverse and well-separated feature clusters in representation learning tasks which is confirmed by our experiments in sections 4.2 and 4.3 on real-world class-imbalance data.
>
> To provide further analysis, we examine the objective function $L_{C_f}(\theta)$ as introduced in Section 3.2 of the paper by separating its formulation $L_{C_f}(\theta) = \sum_{k = 1}^{|C|} f(A_k) - f(\cup_{k = 1}^{|C|} A_k)$ into two parts. The first part resembles the total information function $S_f$ and would enforce intra-class compactness by minimizing the total information in each class $A_k$. Additionally, maximizing $f(\cup_{k = 1}^{|C|} A_k)$ promotes inter-class separation in a quest to maximize the overall diversity in the ground set $\mathcal{V}$. This results in increased inter-class separation.
>
> **W2. Limited Experimental Evidence: The experimental results exhibit certain weaknesses: a) The paper compares its approach with well-known metric learning methods but does not utilize popular metric learning datasets, which could potentially limit the generalizability of the findings. b) All experiments are conducted on relatively small datasets, as opposed to widely recognized datasets commonly used in imbalanced classification, such as ImageNet-LT. This choice of datasets might limit the broader applicability and relevance of the research.**
>
> **A2.** We thank the reviewer for the constructive criticism and pointing us to the corresponding benchmark datasets. We do agree that we were unable to conduct experiments on all possible Longtail benchmarks, especially large scale image datasets due to lack of compute resources.
>
> a. Metric Learning datasets used in approaches like ArcFace(Deng et al., 2019), CosFace (Wang et al., 2018), LiftedStructure Loss (Song et al., 2016) etc., do not demonstrate class-imbalanced settings as demonstrated by real-world data. On the other hand, some of the existing benchmarks in Longtail recognition tasks like ImageNet-LT and LVIS are pathologically created from existing datasets like ImageNet-full and MS-COCO (Lin et al., 2014) respectively. These also do not represent real-world applications like Medical Image Analysis and Autonomous driving. In contrast, the choice of datasets in SCoRe adopt the India Driving Dataset (Varma et al., 2019) which contains approximately 60,000 images of Indian traffic scenes which demonstrate, large-variability among classes, high traffic density (number of objects per image) and natural imbalance (including few-shot objects).
>
> b. We do agree with the reviewer on the choice of relatively smaller datasets due to lack of compute availability, but the datasets chosen in SCoRe represent real-world conditions in medical imaging and autonomous driving which are mission critical applications. For example, the OrganAMNIST and DermaMNIST datasets demonstrate natural imbalance due to variation in modalities and demographic conditions which is prevalent in the real-world as well, thus generalizing to the real-world conditions.
> Nevertheless, the released codebase of SCoRe does have integrations for all possible datasets in metric and longtail learning which the authors will reproduce results in the availability of additional compute as future work.

---

### Official Review · Reviewer_zWRY · 2023-10-31

**Soundness:** 3 good
**Presentation:** 2 fair
**Contribution:** 3 good
**Rating:** 5
**Confidence:** 4

**Summary:**

This paper addresses class imbalance problem in real world for representation learning tasks.
For this purpose, a SCoRe framework and a family of Submodular Combinatorial objectives are proposed to overcome lack of diversity in visual and structural features for rare classes.
Performance evaluation is conducted on two image classification benchmarks (pathologically imbalanced CIFAR-10, subsets of MedMNIST) and a real-world road object detection benchmark (India Driving Dataset ). The newly introduced objectives like Facility Location, Graph-Cut
and Log Determinant can boost the large performance when compared with state-of-the-art metric learners.

**Strengths:**

+ The class-imbalance is a challenging problem, and the illustration of motivation is clear. The effect of class-imbalance on the performance metrics (mAP50) is shown for the object detection task of the IDD.
+ It seems novel by studying metric learners from an assemblage perspective, treating class-specific feature vectors as sets.
+ There are some useful conclusions, e.g., the submodule combinatorial objective can construct more distinguishable clustering features for representation learning. At the same time, the derivation proves that the existing contrastive learning objectives are either submodular or can be reformulated as submodular functions.
+ Three novel objective functions: Facility-Location (FL), Graph-Cut (GC), and Log Determinant (LogDet).
+ Sufficient experiments on datasets with different degrees of class imbalance for different tasks (image classification and image detection), compared to SoTA metric/contrast learners, indicate the importance of combinatorial loss functions.

**Weaknesses:**

- This paper shows comparative analysis related to metric learning and contrastive learning, without focusing on class imbalance issues. Missing some latest methods in Related Work.
- As far as I know, there are various methods available to address class imbalance or long-tail problems, such as focal loss, WPLoss, OHEM, data augmentation... What are the differences between SCoRe and these methods? And there are no comparative experiments with these methods.
- The formulas/symbols in the paper are unclear and lack more explanation.
For instance, 'f' is used to denote both the feature extractor and the submodular function; 'S' is utilized to represent both similarity kernels and total submodular information.
- There are minor writing errors, particularly related to subscript issues, concentrated in Section 3.1. For example, Sij(\theta) , yii=1,2,...|T |.

**Questions:**

- The class imbalance issue may be more pronounced in some other object detection datasets such as the MS COCO[1] or the LVIS[2] which is dedicated to long-tailed object detection. We are looking forward to see some results on them.
- How does SCoRe solve the localization/regression problem in object detection tasks under the class-imbalanced settings?
- Can you provide a detailed explanation of equation (1), as well as the distinction between Total Submodular Information and Total Submodular Correlation?
- Can you provide a visualization of the class distribution in the CIFAR-10 dataset or other dataset?
- Will codes be released in the future?
 [1] Microsoft COCO: Common Objects in Context. ECCV, 2014. [2] LVIS: A Dataset for Large Vocabulary Instance Segmentation. CVPR, 2019.

---

> ### Author Response · Authors · 2023-11-22
>
> **W1. This paper shows comparative analysis related to metric learning and contrastive learning, without focusing on class imbalance issues. Missing some latest methods in Related Work.**
>
> **A1.** We thank the reviewer for bringing this point to light. We do agree that we could not compare against some of the latest methods and have included them as a part of the related work (section 2) section of the main paper. Unfortunately, due to lack of compute resources we are unable to provide the results on all the discussed works.
>
> However, a few points towards the choice of contrastive learners in addressing class-imbalance in SCoRe is noteworthy. Recent works (Suh and Seoh, 2023, Cui et al., 2023 etc.) in longtail learning show that inter-class bias (towards abundant head classes) and intra-class variance are the primary bottlenecks in representation learning tasks. Contrastive / Metric Learners (like SupCon (Khosla et al., 2021)) have been shown to form tighter and well-separated feature clusters (for each class) thereby reducing the impact of inter-class bias and intra-class variance. This makes contrastive learners strong candidates for overcoming class-imbalance. Even SoTA approaches in longtail recognition (Cui et al., 2023, Cui et al. 2022, Zhu et al., 2022 etc.) have adopted contrastive learners in overcoming imbalance alongside additional techniques like data-augmentation, re-balancing etc.
>
> Considering the aforementioned reasons, SCoRe adopts contrastive learning as a premise to **propose a paradigm shift in machine learning by adopting set-based information functions (Submodular functions) as learning objectives and exploiting their combinatorial properties to overcome inter-class bias and intra-class variance.**
> Proposed objective functions (Graph-Cut, Facility Location, Log-Determinant etc.) consider each class in the dataset $\mathcal{T}$ as a set and enforce both intra-class compactness and inter-class separation. For example, the LogDet function introduced in SCoRe minimizes the volume (in the geometric interpretation of LogDet as in (fujishige et al., 2005) of a feature cluster to maintain intra-class compactness while maximizing the volume of the ground set (whole dataset) $\mathcal{V}$ to maximize inter-class separation.
>
> Our experimental results contrasts clearly indicates that submodular combinatorial objectives outperform existing contrastive learners for longtail vision tasks.
>
> **W2. As far as I know, there are various methods available to address class imbalance or long-tail problems, such as focal loss, WPLoss, OHEM, data augmentation... What are the differences between SCoRe and these methods? And there are no comparative experiments with these methods.**
>
> **A2.** We thank the reviewer for pointing this out. We have updated the related work (section 2) of the main paper to highlight the differences between various methods in Longtail learning. Our experiments in the context of longtail object detection (Table 3) compare our proposed objectives (FL and GC) against Focal Loss (Lin et al, 2017) which outperforms OHEM and WPLoss. *Our proposed objectives (as in Faster-RCNN + FPN + FL) outperforms Focal Loss by 23.8 $mAP_{50}$ points*.
>
> We however do not conduct experiments on data-augmentation based architectures as our main focus is to introduce novel objective functions to overcome challenges in longtail settings. Nevertheless our framework facilitates simple integrations of possible backbones and augmentation blocks to further future research in this field.
> We also do not aim to replace the existing SoTA baselines but augment it by using set-based combinatorial objectives in SCoRe.
>
> **W3. The formulas/symbols in the paper are unclear and lack more explanation. For instance, 'f' is used to denote both the feature extractor and the submodular function; 'S' is utilized to represent both similarity kernels and total submodular information.**
>
> **A3.** We thank the reviewer for bringing this inconsistency in the usage of $f$ to light and we have addressed this issue in the updated submission. In the current version the feature extractor is denoted by $F$ while submodular functions have been denoted by $f$.
> However, for the usage of $S$ in our paper, we use $S_{ij}(\theta)$ to denote the similarity between the $i^{th}$ and the $j^{th}$ feature vectors. On the other hand $S_f$ has been used to denote the variant of submodular functions which implements total information (fujishige, 2005). For better clarity, the explanations of symbols and notations have been included in Table 4 in section A.1 of the appendix.
>
> **W4. There are minor writing errors, particularly related to subscript issues, concentrated in Section 3.1. For example, Sij(\theta) , yii=1,2,...|T |.**
>
> **A4.** We thank the reviewer for bringing this inconsistency to light and we have addressed this issue in the updated submission.

---

> > ### Author Response · Authors · 2023-11-22
> >
> > **Q1. The class imbalance issue may be more pronounced in some other object detection datasets such as the MS COCO[1] or the LVIS[2] which is dedicated to long-tailed object detection. We are looking forward to see some results on them.**
> >
> > **A1.** We thank the reviewer for pointing this out. Although we agree that both the datasets (LVIS and COCO) pointed out in the reviews demonstrate extreme class-imbalance. We have updated Section 4.1 with experiments on 1203 classes of the LVIS dataset. Our results in Table 3 show that the Facility Location and Graph-Cut based objective function outperforms the SoTA method (SupCon) by 4.2 $mAP_{50}$ and 2.8 $mAP_{50}$ points respectively on LVIS. A snapshot of the same is given below.
> >
> > |**Method** | **Backbone and head** | $mAP$ | $mAP_{50}$ | $mAP_{75}$  |
> > | ----------- | ----------- | ----------- | ----------- | ----------- |
> > | Faster-RCNN + FPN       | ResNet-101 + FPN  |14.2 | 24.4 | 14.9 |
> > | Faster-RCNN + SupCon | ResNet-101 + FPN  | 14.4 | 26.3 | 14.3 |
> > | Faster-RCNN + Graph-Cut [$C_f$] | ResNet-101 + FPN  | 17.7 | 29.1 | 18.3 |
> > | Faster-RCNN + Facility-Location [$S_f/C_f$] | ResNet-101 + FPN  | **19.1** | **30.5** | **20.3** |
> >
> > Although the LVIS dataset demonstrates extreme class-imbalance, it does not represent real-world unconstrained environments as represented by India Driving Dataset (IDD). The LVIS dataset is created by extending the annotations of COCO 2017 (Lin et al., 2014) dataset which makes it a pathologically curated dataset. On the other hand, the India Driving Dataset (IDD) (Varma et al, 2019) demonstrates large traffic (object) density, large variability between object classes and presence of rare objects (Majee et al., 2021).
> >
> > **Q2. How does SCoRe solve the localization/regression problem in object detection tasks under the class-imbalanced settings?**
> >
> > **A2.** Our experiments on existing object detectors show elevated levels of mis-classification (due to inter-class bias) between abundant and rare objects leading to fall in performance numbers. In most cases however, the bounding box over the target object was predicted accurately.  Thus our framework introduces the submodular objective only to the box classifier in the object detector.
> > Our object detection framework draws inspiration from FSCE (Sun et al., 2021) with proposed modifications to existing Faster-RCNN + FPN based detectors. We have described it below for reference.
> >
> > 1. During the fine-tuning process on Imbalanced datasets we keep the Region Proposal Network (RPN) and the ROI pooling layers unfrozen to adapt to the rare classes.
> > 2. We double the maximum number of proposals kept after Non-Maximal Suppression (NMS), this brings more foreground proposals for rare classes.
> > 3. We consider only half the number of proposals from the ROI pooling layer (top 256 out of 512) for computing the loss function. This forces the objective function to better penalize the object detector for predicting low objectness scores for objects belonging to the rare classes.
> >
> > We have updated the detailed description in the paper. Due to lack of space we have added these details to the Appendix section A.2 of the paper.
> >
> > **Q4. Can you provide a visualization of the class distribution in the CIFAR-10 dataset or other dataset?**
> >
> > **A4.** The class distribution for the CIFAR family of datasets include the (a) balanced (actual dataset) (b) Longtail and (c ) Step based pathological imbalance. Further, DermaMNIST and OrganAMNIST datasets demonstrate natural class imbalance due to varying diversity in various modalities of the captured data. The distributions of all the above datasets had already been included in Figure 3 of the main paper. Finally, the distribution of India Driving Dataset is included in Figure 1 (line plot in blue) of the main paper.
> >
> > **Q5. Will codes be released in the future? [1] Microsoft COCO: Common Objects in Context. ECCV, 2014. [2] LVIS: A Dataset for Large Vocabulary Instance Segmentation. CVPR, 2019.**
> >
> > **A5.** Yes. The code has already been released anonymously (highlighted as a footnote on page 1 of the main paper) at https://anonymous.4open.science/r/SCoRe-8DE5/ for the review process and will be released publicly after the review process. The codebase includes independent modules for each objective function introduced in SCoRe and can be used across classification and detection tasks. Although the code for object detection has not been released as it has been heavy-lifted from the Detectron2 (https://github.com/facebookresearch/detectron2) repository, we have included additional implementation details in section A.2 of the appendix section.

---

> > > ### Author Response · Authors · 2023-11-22
> > >
> > > **Q3. Can you provide a detailed explanation of equation (1), as well as the distinction between Total Submodular Information and Total Submodular Correlation?**
> > >
> > > **A3.** We thank the reviewer for the question. Equation 1 depicts the two variants of submodular information functions as discussed in (Fujishige, 2005). Following the problem definition in SCoRe, each class in the input dataset $\mathcal{T}$ is represented as a set $A_k$ such that the ground-set $\mathcal{V} = A_1 \cup \cdots \cup A_{|C|}$.
> > >
> > > The total information over the ground set $\mathcal{V}$ is denoted as $S_f$, which is calculated as the sum of submodular information measure $f(A_k)$ over each set $A_k \in \mathcal{V}$. Minimizing the total information over each set $A_k$ through an objective function $L(\theta, A_k)$ maximizes the intra-class compactness by minimizing the sum of information over each set $A_k \in \mathcal{V}$, leading to reduced intra-class variance. By the definition of $L(\theta)$ in SCoRe, the $L_{S_f}$ variant of the loss function can be written as $L(\theta) = L_{S_f}(\theta) = \sum_{k = 1}^{|C|} L_{S_f}(\theta, A_k)= S_{f} (A_1,\cdots,A_{|C|}) = \sum_{k = 1}^{|C|} f(A_k)$.
> > >
> > > Similarly, the information correlation of each set $A_k$ with the total information in $\mathcal{V}$ is denoted as the total correlation $C_f$. It is calculated as the difference of the total information measure $S_f$ and the information measure of the ground set $f(\cup_{k = 1}^{|C|} A_k)$. As discussed earlier minimizing the total information $S_f$ leads to reduced intra-class variance, while maximizing $f(\cup_{k = 1}^{|C|} A_k)$ (including the ‘-’ sign in the function) promotes inter-class separation in a quest to maximize the overall diversity in the ground set $\mathcal{V}$. By the definition of $L(\theta)$ in SCoRe, the $L_{C_f}$ variant of the loss function can be written as $L(\theta) = L_{C_f}(\theta) = \sum_{k = 1}^{|C|} L_{C_f}(\theta, A_k)= C_{f} (A_1,\cdots,A_{|C|}) = \sum_{k = 1}^{|C|} f(A_k) - f(\cup_{k = 1}^{|C|} A_k)$.
> > >
> > > In summary, the $S_f$ variant of submodular information functions promotes only intra-class compactness while the $C_f$ variants enforce both intra-class compactness and inter-class separation. This has also been demonstrated in the experiments where the $C_f$ version of the proposed objective functions outperform the $S_f$ variants.

---

### Meta-Review · Area_Chair_qFmb · 2023-12-07

**Metareview:**

Thanks for your submission to ICLR.

This paper considers the class-imbalance problem, and proposes some novel loss functions for contrastive learning.  On the positive side, the reviewers novel that it is an important problem, and the authors did indeed produce some interesting and novel loss functions that perform well in some settings.  On the negative side, the paper is missing some important related work, as well as some key baselines and experiments.  Further, some reviewers felt that the manuscript readability could be improved.

This was clearly a borderline paper.  The author rebuttal addressed some of the concerns, but it still seems that there are some lingering issues, particularly when it comes to additional experiments.  I do also think the paper could use another round of editing to improve readability.  Finally, none of the reviewers indicated that they were willing to argue for accepting the paper, so ultimately it seems that the paper could use some revision and another round of reviewing.

Please do keep in mind the comments of the reviewers when preparing a future version of the manuscript.

**Justification For Why Not Higher Score:**

None of the reviewers were positive enough about the paper, even after the rebuttal, to argue for acceptance or give an accept score.  There seem to be lingering issues with experimental results not covered by the rebuttal.

**Justification For Why Not Lower Score:**

N/A

---

### Decision · Program_Chairs · 2024-01-16

Reject